# MicroRNA-483-5p Inhibits Hepatocellular Carcinoma Cell Proliferation, Cell Steatosis, and Fibrosis by Targeting PPARα and TIMP2

**DOI:** 10.3390/cancers15061715

**Published:** 2023-03-10

**Authors:** Suryakant Niture, Sashi Gadi, Qi Qi, Maxwell Afari Gyamfi, Rency S. Varghese, Leslimar Rios-Colon, Uchechukwu Chimeh, Habtom W. Ressom, Deepak Kumar

**Affiliations:** 1Julius L. Chambers Biomedical Biotechnology Research Institute, North Carolina Central University, Durham, NC 27707, USA; 2The University of Tennessee Health Science Center, Department of Pharmaceutical Sciences, College of Pharmacy, Memphis, TN 38163, USA; 3Lombardi Comprehensive Cancer Center, Georgetown University Medical Center, Washington, DC 20008, USA

**Keywords:** miR-483-5p, NAFLD, AFLD, steatosis, fibrosis, hepatocellular carcinoma

## Abstract

**Simple Summary:**

Hepatocellular carcinoma (HCC) is the fifth leading and highly aggressive lethal liver cancer. The most common cause of HCC is liver cirrhosis because of multiple underlying etiologies, such as chronic hepatitis, nonalcoholic fatty liver disease (NAFLD), alcoholic fatty liver disease (AFLD), and hepatoxicity. In the current study, we characterize the role of microRNA-483-5p in NAFLD/AFLD and HCC progression and its potential use as a prognostic biomarker.

**Abstract:**

MicroRNAs (miRNAs) are small non-coding RNA molecules that bind with the 3′ untranslated regions (UTRs) of genes to regulate expression. Downregulation of miR-483-5p (miR-483) is associated with the progression of hepatocellular carcinoma (HCC). However, the significant roles of miR-483 in nonalcoholic fatty liver disease (NAFLD), alcoholic fatty liver diseases (AFLD), and HCC remain elusive. In the current study, we investigated the biological significance of miR-483 in NAFLD, AFLD, and HCC in vitro and in vivo. The downregulation of miR-483 expression in HCC patients’ tumor samples was associated with Notch 3 upregulation. Overexpression of miR-483 in a human bipotent progenitor liver cell line HepaRG and HCC cells dysregulated Notch signaling, inhibited cell proliferation/migration, induced apoptosis, and increased sensitivity towards antineoplastic agents sorafenib/regorafenib. Interestingly, the inactivation of miR-483 upregulated cell steatosis and fibrosis signaling by modulation of lipogenic and fibrosis gene expression. Mechanistically, miR-483 targets PPARα and TIMP2 gene expression, which leads to the suppression of cell steatosis and fibrosis. The downregulation of miR-483 was observed in mice liver fed with a high-fat diet (HFD) or a standard Lieber-Decarli liquid diet containing 5% alcohol, leading to increased hepatic steatosis/fibrosis. Our data suggest that miR-483 inhibits cell steatosis and fibrogenic signaling and functions as a tumor suppressor in HCC. Therefore, miR-483 may be a novel therapeutic target for NAFLD/AFLD/HCC management in patients with fatty liver diseases and HCC.

## 1. Introduction

Hepatocellular carcinoma (HCC) is the fifth-most common cancer in the world and the third cause of cancer-related mortality [1]. In 2023, the American Cancer Society (ACS) estimated 41,210 new cases of HCC and 29,380 deaths from primary liver cancer and intrahepatic bile duct cancer in the USA (https://www.cancer.org/cancer/liver-cancer/about/what-is-key-statistics.html; accessed on 12 February 2023) [2]. The most common cause of HCC is liver cirrhosis due to various underlying etiologies such as chronic hepatitis B and C viral infection and nonalcoholic fatty liver disease (NAFLD), and alcoholic fatty liver disease (AFLD) [3,4,5,6]. Furthermore, diabetes and obesity epidemics have increased the prevalence of NAFLD and its more severe form, nonalcoholic steatohepatitis (NASH) [7]. Therefore, the risk for HCC, even before the development of cirrhosis, is likely to rise [8,9]. Hepatic steatosis is the earliest event in NAFLD and is characterized by triglyceride accumulation within hepatocytes [10,11]. Ten to twenty percent of steatotic livers develop into NASH, progressing to HCC via fibrosis and cirrhosis [10]. AFLD is another serious public health threat [6,12,13,14,15], and it is estimated that approximately ~5% of the US adult population is affected by this disease. Similar to NAFLD, AFLD progresses histologically defined stages from hepatic steatosis to NASH or alcoholic steatohepatitis (ASH), fibrosis, cirrhosis, and finally to HCC [5,6]. There are various therapeutic approaches for the treatment of HCC, for example, the use of cell therapy, immune checkpoint inhibitors and new tyrosine kinase inhibitors (sorafenib and regorafenib), epigenetic modifiers, and identification and implementation of predictive biomarkers in the treatment of HCC as reviewed recently [16]. However, life expectancy varies depending on the stage of diagnosis, which broadens therapeutic options and improves prognosis [17].

MicroRNAs (miRNAs) are a class of small non-coding RNA molecules (19–25 nucleotides) that binds to 3′ untranslated regions (UTRs) of genes and inhibit gene expression [18]. Several miRNAs have been shown to regulate cell proliferation/survival/apoptosis, cellular metabolism, and stress-related pathways [19,20,21]. Because of their important role in regulating gene expression, miRNAs have been proposed as diagnostic, prognostic, and risk stratification biomarkers in several human cancers, including HCC [22,23,24,25,26,27,28]. Various earlier studies have reported the dysregulation of miR-483 mature variants -3p and -5p across multiple cancer types. For example, elevated expression of miR-483 has been reported in 100% of Wilms’ tumors [29] and also is up-regulated at the hyperplastic stage of pancreatic tumors [30], and approximately 30% of colon, breast, and liver tumors also showed high or even extremely high levels of miR-483-3p expression [29]. The upregulation of miR-483-5p was also associated with poorer disease-specific survival in patients with adrenocortical carcinomas [31], and the expression of miR-483-5p promotes HCC cell proliferation by targeting the suppressor of cytokine signaling 3 (Socs3) [32]. Interestingly, the deregulation of this miRNA by lncRNA might have a role in various cancers. For example, lncRNA TC39A-AS1 acts as a competing endogenous RNA in breast cancer by sponging miR-483-3p, indirectly increasing MTA2 expression and tumorigenicity of breast cancer. [16]. Similarly, in tongue squamous cell carcinoma (TSCC) lncRNA NR_034085, miRNA processing–related lncRNA (MPRL) directly binds to pre-miR-483 within the loop region and blocks pre-miR-483 recognition and cleavage by TRBP–DICER-complex, thereby inhibiting miR-483-5p generation which leads to upregulation miR-483-5p downstream target-FIS1 expression [33]. These studies suggest that miR-483 acts as an oncogenic micro-RNA in several cancers.

Given the complexity of the mechanisms behind NAFLD, AFLD, and HCC development, there remains a gap in the knowledge regarding the role of crucial miRNAs and their downstream gene targets in transitioning from a healthy liver to AFLD/ALD-NASH-HCC. In the current study, we examined the association between miR-483-5p (miR-483) expression in NAFLD/AFLD in vivo mice models and HCC patient tissues to better understand the role of this miRNA in liver diseases. Reports suggest a possible role of miR-483 in liver disease. MiR-483-5p targeted the proprotein convertase subtilisin/kexin type 9 (PCSK9) 3′-UTR, leading to decreased PCSK9 protein and mRNA expression, increased hepatic LDL receptor expression, and enhanced LDLcholesterol uptake [34]. Overexpression of miR-483 in mice liver increased hepatic LDL receptor levels by targeting PCSK9, leading to decreased plasma total cholesterol and LDL cholesterol levels, suggesting that microRNA-483 ameliorates hypercholesterolemia [34]. An earlier study demonstrated that miR-483 targets metalloproteinase 2 (TIMP2) and platelet-derived growth factor-β (PDGF-β), thus suppressing CCl_4_-mediated mouse liver fibrosis in vivo [35].

Our data suggest that miR-483 overexpression inhibited cell proliferation/migration, induced apoptosis, and dysregulated Notch signaling. Overexpression of miR-483 inhibited cell steatosis and downregulated fibrogenic signaling by targeting peroxisome proliferator-activated receptor alpha (PPARa) and tissue inhibitor of metalloproteinases 2 (TIMP2), respectively. We also found downregulation of miR-483-5p in both mice models of NAFLD and AFLD and human HCC tissue samples.

## 2. Materials and Methods

### 2.1. HCC Tumor Samples and miR-483 Expression

Patients were recruited at MedStar Georgetown University Hospital (MGUH), Washington, DC. The study is conducted through a protocol approved by the Georgetown-Howard Universities Center for Clinical and Translational Science (GHUCCTS) Institutional Review Board (IRB), Washington, DC, under Protocol #2014–0059 “Multi-omic Approaches for Liver Cancer Biomarker Discovery.” The collection and use of the tissues was approved by the IRB of Georgetown University, Washington, DC, under Protocol #2007–345 “Establishment of the High-Quality Tumor Biobank and Clinical Database”. All patients signed a consent form permitting the use of donated tissue. The consent forms and their content were reviewed and approved by the IRB. Detailed characteristics of the study cohort are described in Appendix A. In this study, we analyzed 80 samples consisting of 40 tumor tissues (HCC) and 30 adjacent normal tissues (Adj-N), and 10 adjacent cirrhotic tissues (Adj-C) acquired from 40 HCC patients, including 14 African American (AA), 16 European American (EA), and 10 Asian American (AAM). We excluded from analysis 1 sample from an AA patient and one sample from an Asian patient due to outlier screening of the corresponding mRNA-Seq data.

Total RNAs were isolated from samples as previously described [36]. Briefly, RNA samples were isolated using the RNeasy Plus Universal Mini Kit (Qiagen, Hilden, Germany) following the manufacturer’s instructions. The quality and concentration of RNA were estimated using the NanoDrop ND-1000 spectrophotometer. Further analysis of RNA integrity was performed using the Agilent RNA 6000 Nano Kit on the Agilent 2100 Bioanalyzer. Libraries were prepared using the TruSeq RNA Access Library Prep Kit. Sequencing was performed in an Illumina HiSeq 4000 instrument using a 150 bp pair-end (PE150). The mRNA-Seq data contained an average of 33 M reads per sample. The fastq files were then imported into Partek Flow for quality assessment, alignment, and estimating transcript abundance. Alignment was performed using the spliced transcripts alignment (STAR) algorithm, and reads were quantified using the Expectation Maximization (E/M) method implemented in Partek Flow with Trimmed Mean of M-values (TMM) used for normalization. The miRNA-Seq data were analyzed using the QIAseq miRNA quantification data analysis software (https://geneglobe.qiagen.com/us/analyze, accessed on 12 February 2023). The first step is the primary analysis, where the unique molecular index (UMI) counts are calculated, and primary miRNA mapping is performed. In the secondary analysis step, the UMI counts are analyzed to calculate the changes in miRNA expression. The quantified data were then normalized using the TMM method before any statistical analysis was performed. All raw and pre-processed miRNA-seq, mRNA-seq, and DNA methylation data have been deposited in NCBI’s Gene Expression Omnibus (GEO) and are accessible through GEO Series accession number GSE176289 (https://www.ncbi.nlm.nih.gov/geo/query/acc.cgi?acc=GSE176289, accessed on 12 February 2023), as indicated [37].

### 2.2. Cell Culture

Human HCC HepG2 (cat #HB-8065), SK-Hep1(cat #HTB-52), and Hep3B (Cat #HB-8064) cells were obtained from American Type Culture Collection (ATCC). HCC cells were grown in DMEM medium (Invitrogen, Carlsbad, CA, USA) supplemented with 5% Fetal Bovine Serum (FBS, Access Biologicals, Vista, CA, USA) and 50 U/mL penicillin/streptomycin. HepaRG, a human bipotent progenitor cell line capable of differentiating into 2 different cell phenotypes (i.e., biliary-like and hepatocyte-like cells) [38], was obtained from ThermoFisher Scientific (Waltham, MA, USA). The terminally differentiated HepaRG cells (Cat #HPRGC10) and media ingredients were obtained from ThermoFisher Scientific (Waltham, MA, USA). As per the manufacturer’s instructions, specific thawing and plating media (Cat #HPRG770) were used, and cells were expanded in William’s E Medium (Cat #12551032) supplemented with 1% GlutaMax (Cat #35050061). Human cryopreserved hepatocytes (HPCH05+) and hepatocytes thawing and plating medium were obtained from Xenotech (Kansas City, KS, USA). Cells were incubated at 37 °C in a cell culture incubator supplied with 5% CO_2_ and used in experiments when they reached 70–80% of the confluence level.

### 2.3. miRNA Transfection

HepaRG, HepG2, SK-Hep1, Hep3B cells, and human hepatocytes were grown in 6-well plates (1 × 10^5^ cells/wells) for 24 h before transfection. Cells were transfected with mirVana miRNA-483-5p mimic (miR-483) (AAGACGGGAGGAAAGAAGGGAG; Cat #4464066), mirVana miRNA-483-5p inhibitor (miR-483 Inh.) (AAGACGGGAGGAAAGAAGGGAG; Cat #4464084) or mirVana miRNA Mimic Negative Control #1 (NC) (Cat #4464058). All mirVana miRNA mimics, negative control, and miRNA inhibitor were obtained from Ambion (Austin, TX). HCC cells and hepatocytes were transfected with 100 nM of negative control (NC) or 25–100 nM of miR-483 mimic or miR-483 inhibitor using the Lipofectamine-2000 reagent (Invitrogen, Waltham, MA, USA). Cells were harvested 30 h after transfection. The expression of miR-483 target genes was analyzed by RT/qPCR, and protein expression was assessed by immunoblotting.

### 2.4. RT/qPCR

Total RNAs from HepaRG, HepG2, SK-Hep1, and Hep3B cells were isolated using TRIZOL reagent (Life Technologies, Carlsbad, CA, USA). In other experiments, HepaRG, HepG2, SK-Hep1, Hep3B cells, and human hepatocytes were transfected with NC mimic or miR-483 mimic separately for 30 h. Cells were washed with PBS, and total RNAs were isolated using TRIZOL. Equal amounts of RNA (1 µg) were reverse transcribed using the High-Capacity cDNA Reverse Transcription kit (Applied Biosystems, Waltham, MA, USA). Then, cDNA was incubated with Power SYBR Green PCR master mix (Applied Biosystems) with appropriate forward and reverse primers of indicated genes (Appendix A). GAPDH was used as an internal control. 

For miR-483 expression analysis, total miRNAs from HepaRG, HepG2, SK-Hep1, Hep3B cells, and human hepatocytes were isolated using the mirVana microRNA Isolation Kit (Thermo Fisher Scientific, Waltham, MA, USA). The total miRNAs (10 ng) were reverse transcribed using primers specific for miR-483 and RNU44 (Assay ID 002338 and 001094, respectively, Applied Biosystems, Carlsbad, CA, USA) and TaqMan Reverse Transcription reagents (Applied Biosystems). Expression of miR-483 and RNU44 was quantified by RT/qPCR using TaqMan PCR master mixture and Taqman expression assay primers. NU44 expression was used as an internal control. To quantify miR-483 expression in mice liver, we utilized primers specific for miR-483 and snoRNA202 (Assay ID 001232; as an endogenous control). All PCR reactions were performed on a QuantStudio-3 PCR system (Applied Biosystems), and relative quantitation was analyzed according to the manufacturer’s protocols.

### 2.5. Western Blotting

Immunoblotting was performed as described previously [39]. Briefly, after transfection with miR-483 or NC mimics/inhibitors, HCC cells were lysed in cell lysis buffer (Cell Signaling Technology, Danvers, MA, USA) containing a protease inhibitor cocktail (Roche, Indianapolis, IN). After centrifugation at 10,000 RPM for 15 min, the cell lysate supernatants were used for protein qualification. Protein concentrations were measured using the Bio-Rad protein assay reagent (Bio-Rad, Hercules, CA, USA). Sixty micrograms of cell lysates were electrophoresed by using NuPAGE 4–12% Bis-Tris-SDS gels (Invitrogen, Waltham, MA, USA), and proteins were then transferred to polyvinylidene difluoride (PVDF) membranes (Millipore, Billerica, MA, USA). After washing the membranes with 1× Tris-buffered saline with 0.1% Tween 20 Detergent (TBS-T), the membranes were blocked in 1× blocking buffer (Sigma-Aldrich, St. Louis, MO, USA) for 1 h. The membranes were then incubated with primary antibodies overnight at 4 °C as per the manufacturer’s protocols. The following antibodies were obtained from Cell Signaling Technology (Danvers, MA, USA): anti-Notch1 (Cat #4147s), anti-Notch2 (Cat #5132s), anti-Notch3 (Cat #5276s), anti-Hes1 (Cat #11988s), anti-cleaved-PARP (Cat #9541S), anti-LC3B (Cat #4108S), anti-p62 (Cat #5114s), anti-GAPDH (Cat #5174S), anti-β-actin (Cat #4970S), anti-E-cadherin (Cat #3195S), anti-N-cadherin (Cat #13116S), anti-Vimentin (Cat#5741S), anti-Nanog (Cat #4903s), anti-p21(Cat #2947s), anti-CD44 (Cat #3570s), anti-TIMP2 (Cat #5738s), anti-MMP2 (Cat #13132s), anti-TGFβ (Cat #3711s), anti-fatty-acid synthase (FASN) (Cat #3180S), anti-SCD1 (Cat #2794S), and anti-ACC (Cat #3662S). We also obtained anti-L-FABP (Cat #ab7366) from Abcam, anti-TNFAIP8 (Cat #15790-1-AP) antibody, and anti-PPARA (Cat #15540-1-AP) from Proteintech (Rosemont, IL, USA); anti-PPRA-γ (Cat #sc-7273) and anti-SREBP1 (Cat #sc-13551) from Santa Cruz Biotechnology (Dallas, TX, USA). After overnight incubation, the membranes were washed 3 times with TBST and then incubated in the appropriate secondary antibody (1:10,000 dilution) (Jackson ImmunoResearch, West Grove, PA, USA) for 1 h at room temperature. The immunoreactive bands were visualized using Enhanced chemiluminescence (ECL) detection reagents (Signagen Laboratories, Rockville, MD, USA). The immunoblots were visualized using the Azure C-500 Bio-system.

### 2.6. Cell Survival Assay

Cells (1 × 10^4^ cells/well) were grown in 96 plates and transfected with NC (100 nM) or increasing concentrations of miR-483 mimic (25–100 nM) for 72 h. In a separate experiment, cells transfected with NC or miR-483 were also treated with sorafenib (5µM), regorafenib (2.5 µM), or their corresponding vehicle controls and incubated for 72 hrs. Cells were then incubated with 5 µL/well of MTT (3-(4,5-dimethylthiazol-2-yl)-2,5-diphenyltetrazolium bromide) reagent (5 mg/mL) for 1 h at 37 °C in a cell culture incubator. Cells were then carefully washed with PBS, and formazan crystals were dissolved in 100 µL DMSO. Cell survival was determined by quantifying absorbance at 570 nm using a Fluostar Omega plate reader (BMG Lab tech, Cary, NC, USA). All experiments were repeated 3 times.

### 2.7. Cell Colony Formation Assay

HepaRG, HepG2, and SK-Hep1 cells (1 × 10^5^ cells/well) were grown in 6-well plates in triplicates for 18 h and transfected with NC (100 nM) or miR-483 (100 nM) for 24 h. After transfection, cells were trypsinized and counted, and live cells (5000 cells/well) were re-plated in 6-well plates in triplicates. Cells were allowed to grow for 7–10 days until colonies were visible. Cell colonies were then washed with PBS for 1 min, fixed with cold methanol, and stained with 0.1% crystal violet for 1 h. Cell colonies were washed with distilled water and allowed to dry. Finally, cell colonies were photographed, counted, and plotted.

### 2.8. Cell Migration Assay

The effect of miR-483 mimic or miR483 inhibitor (miR-483 Inh.) on the migratory ability of HepaRG and SK-Hep1 cells was determined by wound healing migration assay as described previously [39,40]. Cells (1 × 10^6^ cells/well) were grown in a 6-well plate for 18 h and transfected with 100 nM of NC, miR-483 mimic, or miR-483 inhibitor mimic for 24 h. After transfection, a cell monolayer was scraped using a micropipette tip (A_0_). At 24 h post-wounding (A_24_), cells were photographed, and the migration gap length was calculated using ImageJ software v.1.8.0 (https://imagej.nih.gov/ij/, accessed on 12 February 2023). The percent wound closure was calculated using the formula [(A_0_ − A_24_)/A_0_] × 100 and plotted.

### 2.9. Luciferase Assay

HepG2 and SK-Hep1 cells (1 × 10^4^ cells/well) were transfected with 0.5 µg of TIMP2-3′UTR-Luciferase construct or PPARA-3′UTR-Luciferase (OriGene, custom designed) in 6-well plates for 18 h. Cells were then transfected with 100 nM of NC and miR-483 mimic or miR-483 mutant mimics (custom-designed from Integrated DNA Technologies) for 24 h. Transfected cells were washed with PBS, lysed, and 20 µg lysates were mixed with luciferase substrate (Promega, Madison, WI, USA). Plates were covered with aluminum foil to protect them from light and incubated at room temperature for 20 min. Fluorescence was measured using a Fluostar Omega plate reader (BMG Lab Tech, Cary, NC, USA), and relative luciferase activity was measured and plotted.

### 2.10. Development of NAFLD and AFLD Mouse Models

All animal handling and procedures were carried out as per NIH Guidelines for the Care and Use of Laboratory Animals and approved by the NCCU-Institutional Animal Care and Use Committee (NCCU-IACUC Protocol No. is MG-02-26-2010). For the NAFLD development model, 10-week-old male C57BL/6J mice were fed a regular chow diet (control diet, 12% calories as fat; *n* = 5) or a high-fat diet (HFD, 45% calories as fat; *n* = 5) for 16 weeks. For the AFLD model, 10-week-old male C57BL/6J mice were ear tagged and randomly assigned to one of 2 groups and either pair-fed a control diet (*n* = 5) or a standard Lieber-Decarli liquid diet containing 5% EtOH (*n* = 5) (representing 27.5% of the total caloric intake), for 8 weeks as previously described [41,42]. Liquid diets, purchased from DYETS Inc (Bethlehem, PA, USA), were based upon the Lieber-DeCarli EtOH formulation and provided 1 kcal/mL. Our pre-established inclusion/exclusion criteria were that animals would be excluded from the analysis if they were too sick or died before the end of the study. After 16 weeks of high-fat diet feeding or after 8 weeks of a standard Lieber-Decarli liquid diet, mice were anesthetized with isoflurane and sacrificed. Livers were isolated, weighed, and sections were rapidly dissected, snap-frozen in liquid nitrogen, and kept at −80 °C. A part of the fresh liver tissues was fixed in 10% formalin liver slices prepared using a cryostat. The sections were stained with hematoxylin and eosin (H&E) staining for histological examination, as described previously [42]. Total RNAs from liver tissues were isolated and purified, and the expression of miR-483, fibrosis markers, or notch signaling gene expression was analyzed by RT/qPCR.

### 2.11. Statistical Analysis

All experiments were performed in triplicates and presented as mean ± SEM. Differences between groups were analyzed using a 2-tailed Student’s *t*-test. A *p*-value of < 0.05 was considered statistically significant. Statistical significance was determined by Graph Pad Prism 9 software (GraphPad Software Inc., La Jolla, CA, USA).

## 3. Results

### 3.1. Downregulation of miR-483 Expression Activates Notch Signaling in HCC Tissues

To investigate the biological significance of miR-483 in HCC, we first analyzed the expression levels of miR-483-5p and miR483-3p in liver hepatocellular carcinoma (LIHC) as reported in The Cancer Genome Atlas (TCGA) data set using the MIR-TV portal (http://mirtv.ibms.sinica.edu.tw/analysis.php; accessed on 9 August 2022). Analysis of the available TCGA data suggests a significant downregulation of both miR-483-5p (*n* = 345; *p* < 0.001) and miR-483-3p (*n* = 364; *p* < 0.001) expression in HCC tumors compared with normal liver tissues (*n* = 50) (Figure 1A, upper and lower panels). To support this observation, we analyzed the expression of miR-483-5p and miR-483-3p in HCC tumor tissues from African American (AA), European American (EA), and Asian American (AAM) patients as described in the method section (Figure 1B). The expression of miR-483-5p and miR-483-3p were significantly (*p* < 0.001) down-regulated in HCC tissues from AA (*n* = 13) and EA (*n* = 16) patients but not in AAM patients (*n* = 9) compared with normal liver tissues (Figure 1B, upper and middle panels). Since numerous reports suggest that the Notch 3 receptor is constitutively active in HCC [43,44], we also analyzed its expression in our sample cohort. Our data suggest that *Notch 3* gene expression was significantly (*p* < 0.05) higher in the HCC tissues of AA, EA, and AAM patients compared with matched normal liver tissue samples (Figure 1B, lower panel).

We then asked whether the miR-483-5p (hereafter called miR-483) expression is co-related with the Notch 3 expression. We analyzed miR-483 and Notch 3 expression in HepaRG (a human bipotent progenitor liver cell line) and HCC cell lines HepG2 and SK-Hep1. RT/qPCR data suggest that miR-483 endogenous expression was not significantly changed in HepG2 as compared with HepaRG cells, but a significantly higher expression was observed in SK-Hep1 cells (Figure 1C, upper panel). However, *Notch 3* expression was significantly higher in HCC HepG2 and SK-Hep1cells compared with HepaRG cells (Figure 1C, lower panel).

Immunoblotting data suggest that the expression of Notch 3 protein was downregulated in SK-Hep1 cells compared with HepaRG or HepG2 cells. No change in Notch 2 and Notch 1 expression between HepaRG, HepG2, and SK-Hep1 was observed. Interestingly, the Notch downstream target Hes1 was downregulated in SK-Hep1 cells, whereas miR-483 expression was significantly upregulated (Figure 1C (upper panel),D). Our overall data suggest that the expression of miR-483 is downregulated in HCC tumors, and expression of miR-483 may affect Notch(s) signaling in HCC tumors and HCC cell lines.

### 3.2. Overexpression of miR-483 Dysregulates Notch Signaling in HCC Cells 

Notch signaling plays various roles in HCC by regulating tumorigenesis, angiogenesis, invasion, and metastasis [45]. Increased Notch expression has also been associated with poor prognosis in HCC patients [45,46]. In addition, our data demonstrated that miR-483 expression is downregulated in HCC patients’ tumor tissues, and downregulation of miR-483 is associated with the upregulation of Notch 3. To further explore an association between the regulation of miR-483 and Notch(s) signaling in HCC, we analyzed the impact of miR-483 overexpression on Notch signaling in HepaRG and HCC cells. HepaRG and HepG2, and SK-Hep1 cells were transfected with negative control (NC) or miR-483 mimic, and the overexpression of miR-483 was analyzed by RT/qPCR (Figure 2A). Under similar experimental conditions, we also analyzed the expression of *Notch 1*, *Notch 2*, *Notch 3* and *Notch 4* genes by RT/qPCR (Figure 2B). The overexpression of miR-483 increased the expression of *Notch 1* and *Notch 3* and decreased the expression of *Notch 2* and *Notch 4* in HepaRG cells (Figure 2B). Interestingly, overexpression of miR-483 significantly reduced *Notch 3* expression in HepG2 and SK-Hep1 cells (Figure 2C, left and right panels). Also, the Notch downstream target *HES 1* expression was downregulated in SK-Hep1 (Figure 2C, right panel). No significant change in *Notch 4* expression was observed when miR-483 was overexpressed in HepaRG, HepG2, and SK-Hep1 cells (Figure 2A–C).

Further, we analyzed the effect of miR-483 overexpression on full-length Notch 3, Notch 2, and Notch 1 proteins, the cleaved Notch (s) transmembrane/intracellular (NTM) fragments, and Notch downstream Hes1 protein expression by immunoblotting (Figure 2D, left and right panels). Transient transfection of miR-483 mimic (50 and 100 nM) decreased expression of full-length Notch 3, Notch 2, and Notch 1 and NTM cleaved fragments in HepaRG and HepG2 cells. Full-length Notch 3 and NTM regions expression also decreased in SK-Hep1 cells after overexpression of miR-483. Hes1 expression decreased in HCC HepG2 and SK-Hep1 cells after overexpression of miR-483 (Figure 2D, left and right panels). On the other hand, the inactivation of miR-483 by miR-483 inhibitor stabilized full-length Notch 3, Notch 2, and NTM regions, and no effect on HES1 expression was observed in our cellular models (Figure 2E). Collectively, our data suggest that miR-483 affects/downregulates Notch signaling in HCC cells. However, the underlying molecular mechanisms need to be further investigated.

### 3.3. miR-483 Inhibits HCC Hallmarks and Increases Sensitivity toward Anti-HCC Drugs

Since the overexpression of miR-483 downregulates Notch signaling in HCC cells, we further analyzed the impact of miR-483 on HCC cell survival, colony formation ability, migration, and epithelial-mesenchymal transition (EMT). Our data demonstrate that dose-dependent overexpression of miR-483 inhibits cell survival in HepaRG (72%), HepG2 (14%), SK-Hep1 (63%), and Hep3B (14%) compared with NC (100 nM) transfected cells (Figure 3A). Cell colony formation assay showed that miR-483 inhibits HepaRG (61%), HepG2 (54%), and SK-Hep1 (56%) cell colony formation (Figure 3B, upper and lower panels).

EMT is essential in the transition from localized disease to invasion and metastasis in cancers [47] and since EMT plays an important role in cell migration and invasion in HCC [48]. Here we analyzed the effect of miR-483 on cell migration using a cell scratch assay (Figure 3C). We observed a significant reduction in cell migration in HepaRG by ~70% and SK-Hep1 cells by ~74% transfected with miR-483 mimics compared with NC transfected cells (Figure 3C). Inhibiting miR-483 did not have any significant effects, compared to NC-transfected cells (Figure 3C, left and right panels). In addition, we studied the changes in the expression of various EMT markers when the miR-483 expression is modulated (Figure 3D). Our immunoblotting data suggest that expression of miR-483 increased E-cadherin and decreased N-cadherin expression in HepaRG and HepG2 cells. In SK-Hep1 cells, miR-483 expression increased E-cadherin, N-cadherin, and decreased Vimentin. Inactivation of miR-483 using an inhibitor increased Vimentin in all three cell lines studied, suggesting that miR-483 suppresses EMT in HepaRG and HCC cells and inactivation of endogenous miR-483 promotes EMT in a cell dependent manner (Figure 3D). Since miR-483 inhibits cell survival and migration, we also analyzed the effect of miR-483 on the expression of Nanog and CD44 cancer stem cell markers and oncogenic TNFα-Induced Protein 8 (TNFAIP8) marker. Overexpression of miR-483 decreased the expression of Nanog, CD44, and TNFAIP8 in HepaRG and CD44 and TNFAIP8 in HepG2 cells (Figure 3E). The expression of Nanog, CD44, and TNFAIP8 in SK-Hep1 also decreased after overexpression of miR-483 (Figure 3E), suggesting that miR-483 inhibits HCC progression.

To further explore the role of miR-483 in cell apoptosis, we assessed cleaved-PARP (c-PARP) expression in HepaRG, HepG2, and SK-Hep1 cells after transfection with miR-483 mimic and NC. Overexpression of miR-483 increased cleaved-PARP (c-PARP) expression in HepaRG, HepG2, and SK-Hep1 cells compared with NC transfected cells (Figure 3F). Inactivation of miR-483 decreased cleaved-PARP (c-PARP) expression in all three cell lines, suggesting that miR-483 expression induced cell apoptosis (Figure 3F). Since our data suggest that miR-483 potentially suppresses HCC hallmarks and induces cell apoptosis, we further examined the potential therapeutic role of miR-483 in combination with anti-neoplastic drugs (sorafenib and regorafenib). As expected, expression of miR-483 decreased cell survival in HepaRG and HepG2, SK-Hep1, and Hep3B HCC cells (Figure 3G, left and right panels). Exposure of sorafenib (5 µM) or regorafenib (2.5 µM) alone decreased cell survival in all cell lines. Interestingly, sorafenib (5 µM) or regorafenib (2.5 µM) combined with miR-483 expression further reduced cell survival in all cell lines compared with NC transfected and sorafenib or regorafenib-only treated cells (Figure 3G). These results suggest that miR-483 could have an additive or synergistic effect potentiating drug sensitivity in HCC. Indeed, our data indicate that miR-483 suppressed cancer hallmarks in HCC and could be a potential biomarker for this disease.

### 3.4. miR-483 Inhibits HCC Cell Steatosis by Modulation of Lipogenic Gene Expression

NAFLD is a major risk factor for the development of HCC in non-cirrhotic patients, and the expression of lipogenic enzymes/proteins modulates hepatic steatosis and NAFLD development [49]. Since miR-483 inhibits HCC cell proliferation, we next address the impact of miR-483 on steatosis, an early event in NAFLD development. We transfected human hepatocytes and HepaRG, HepG2, SK-Hep1, and Hep3B cells with miR-483 mimic (50 and 100 nM). We then evaluated the effects of miR-483 on the regulation of genes involved in lipogenesis, such as Acetyl-CoA carboxylase (*ACC*), liver fatty acid-binding protein-1 (*L-FABP1*), fatty acid synthase (*FASN*), stearoyl-CoA desaturase-1 (*SCD1*), transcription factors such as sterol regulatory element-binding protein 1 (*SREBP1*), and peroxisome proliferator-activated receptor alpha and gamma (*PPARG*). Overexpression of miR-483 in human hepatocytes slightly decreased *FASN* and *SCD1* expression, but no significant changes in *ACC* and *PPARG* were observed. On the other hand, increased *FABP1* and *SREBP1* expressions were observed compared with NC transfected cells (Appendix A). Similarly, *SREBP1*, *PPARG,* and *FABP* expression were significantly decreased in HepaRG cells, *SCD1* in SK-Hep1 cells, and *SREBP1* in Hep3B cells. At the same time, the overexpression of miR-483 increased *FASN* and *ACC* in HepaRG, HepG2, and SK-Hep1 cells compared with NC transfected cells (Appendix A), suggesting that miR-483 dysregulates lipogenic signaling in hepatocytes and HCC cells.

On the contrary, we inactivated endogenous miR-483 by using a miR-483 inhibitor, and lipogenic gene expression was analyzed. Inactivation of endogenous miR-483 expression increased the expression of *SCD1*, *FASN*, *PPARG*, and *SREBP1* in HepaRG cells, *ACC*, *SCD1,* and *FASN* in HepG2 cells, and *FASN* in SK-Hep1 cells compared with NC transfected cells (Figure 4A). These results suggest that miR-483 has a role in the modulation of lipogenic gene expression. We then analyzed the expression of miR-483 in HepaRG and HepG2 after exposure of the cells to fatty acids such as oleic acid (OA), elaidic acid (EA), palmitic acid (PA), lauric acid (LA), stearic acid (SA), myristic acid (MA), linoleic acid (LNA), and cholesterol (CHO) which are known to induce cell steatosis. Treatments with OA, LA, CHO, PA, MA, SA, and LNA increased the expression of miR-483 in HepaRG cells compared to untreated/control (Figure 4B, upper panel). Similarly, OA, LA, CHO, EA, and MA significantly increased miR-483 expression in HepG2 cells (Figure 4B, lower panel), indicating that exposure to fatty acid/cholesterol increased endogenous miR-483 expression in HepaRG and HepG2 cells.

Since our data suggest that overexpression or inactivation of miR-483 dysregulates lipogenic signaling, we then examined the role of miR-483 on cell steatosis after transfecting HepaRG, HepG2, SK-Hep1, and Hep3B cells with miR-483 mimic (50 and 100 nM) or miR-483 inhibitor (100 nM) compared to their corresponding NC (100 nM) control. The cells were first transfected with NC mimic, miR-483 mimic, or miR-483 inhibitor for 24 h and exposed to OA for an additional 24 h, and cell steatosis was examined using Oil Red O (ORO) staining (Figure 4C). Overexpression of miR-483 (50–100 nM) significantly decreased lipid droplet accumulation in HepaRG and HCC cells. In contrast, inhibition of miR-483 restored/promoted cell steatosis (Figure 4C). We quantified steatosis after ORO staining as described in the methods section, and our data suggest that miR-483 inhibits cell steatosis in HepaRG, HepG2, SK-Hep1, and Hep3B cells, whereas inactivation of miR-483 promotes cell steatosis significantly (Figure 4D, upper and lower panels). Furthermore, immunoblotting data suggest that overexpression of miR-483 decreased expression of SCD1, FASN, PPARγ, L-FABP, and SREBP1 in HepaRG cells and SCD1, L-FABP, and SREBP1 in HepG2 cells (Figure 4E). Increased expression of FASN, ACC, and PPARG was also observed in HepG2 cells transfected with miR-483 (Figure 4E), suggesting that miR-483 modulates the expression of steatosis/lipogenesis markers that leads to inhibition of cell steatosis.

Autophagy plays an essential role in lipid metabolism/lipid droplet clearance [50]. Since overexpression of miR-483 inhibits cell steatosis, we analyzed the effect of miR-483 on the expression of known autophagy biomarkers such as LC3B I/II and p62 in HCC cells (Figure 4F). Overexpression of miR-483 in HepaRG, and HCC HepG2, SK-Hep1, and Hep3B cells increased LC3B I/II and decreased p62 compared with NC transfected cells (Figure 4F), suggesting that miR-483 could inhibit cell steatosis by inducing autophagy. Our data suggest that miR-483 dysregulated lipogenic gene expression and suppressed cell steatosis by activating cellular autophagy.

### 3.5. miR-483 Inhibits Fibrogenic Signaling in HCC

NAFLD/AFLD progresses through histologically defined stages from hepatic steatosis to steatohepatitis (NASH), fibrosis, cirrhosis, to HCC. To investigate the role of miR-483 in fibrosis, we inactivated miR-483 in HepaRG and HCC hepG2 and SK-Hep1 cells (Figure 5A). The inactivation of endogenous miR-483 expression by miR-483 inhibitor increased expression of *TGFβ*, tissue inhibitor of metalloproteinase 2 (*TIMP2*) and *p21* in HepaRG cells, *TGFβ*, *TIMP2*, *p21, Cytokeratin 7* in HepG2 cells and *p21* in SK-Hep1 cells several folds compared with NC transfected cells (Figure 5A) suggesting that miR-483 can affect the expression of *p21*, *TGFβ*, *TIMP2*, and *Cytokeratin 7*. As earlier reported, miR-483 targets *TGFβ* [51], and our data indicate that miR-483 overexpression dysregulated TGFβ expression and decreased TIMP2 and matrix metallopeptidase 2 (MMP2) gene expression. We validated these results by immunoblotting using HepaRG and HepG2 cells. The overexpression of miR-483 decreased TIMP2 and TGFβ expression in HepaRG and HepG2 cells and MMP2 in HepG2 cells compared to NC-transfected cells (Figure 5B).

Interestingly, when HepaRG and HepG2 cells were exposed to carbon tetrachloride (CCl_4_), a known fibrosis inducer agent in liver cells, the increased expression of miR-483 was observed in HepG2 cells and significantly decreased expression of miR-483 was observed in HepaRG cells compared with untreated cells (Figure 5C, upper panel). Immunoblotting data demonstrated that overexpression of miR-483 and CCl_4_ (10 mM) treatments decreased TIMP2 and TGFβ protein expression in HepG2 cells and MMP2 and TIMP2 in HepaRG cells (Figure 5D, left and right panels). These results suggest that CCl_4_ exposure could modulate miR-483 expression affecting downstream targets TIMP2 and TGFβ expression. Taken together, our data suggest that miR-483 downregulates fibrogenic signaling in HCC cells.

### 3.6. miR-483 Targets PPARa and TIMP2 and Inhibits Cell Steatosis and Fibrosis

To further understand the molecular mechanisms of how miR-483 inhibits cell steatosis and fibrogenic signaling in HCC cells, we used TargetScan (http://www.targetscan.org/vert_72/; accessed on 12 August 2022) to identify possible 3′-untranslated regions (UTR) of genes targeted by miR-483. TargetScan analysis revealed that miR-483 binds to the UTRs of the *PPARA* gene (*PPARA* isoform 5; 2638–2644), *TIMP2* (2089–2095), and *p21* (705–711) (Figure 6A). Interestingly, peroxisome proliferator-activated receptors (PPARs) play a critical role in lipid metabolism and homeostasis in multiple cell types [52,53]. Also, an earlier report demonstrated that miR-483 targets TIMP2 and PDGF-β leads to suppressed CCl_4_-mediated liver fibrosis in mice [35], and miR-483 also targets *TGFβ* which is involved in fibrogenic responses in pulmonary arterial hypertension (PAH) [51]. The binding sequences of miR-483 with these genes’ UTRs are presented in Figure 6A. The binding of miR-483 to the 3′ UTR of *TIMP2* and *PPARa* genes was analyzed by luciferase reporter assay. Co-transfection of 3′ UTR of *TIMP2* and *PPARA* gene and wild-type miR-483 decreased luciferase activity significantly (Figure 6B, left panel), but no significant change in luciferase activity was observed in mutant-type miR-483 transfected cells (Figure 6A,B (right panel)).

To further address the molecular mechanisms of how miR-483 regulates the expression of these genes, we first analyzed the effect of miR-483 on the regulation of *PPARA*, *TMIP2,* and *p21* gene expression and protein expression in HepaRG and two HCC HepG2 and SK-Hep1 cell lines. RT/qPCR data suggest that overexpression of miR-483 decreased *p21, PPARA(5),* and *TIMP2* expression significantly in HepaRG cells and *PPARA(5)* and *TIMP2* gene expression in HCC cell lines (Figure 6C). Immunoblotting data suggest that overexpression of miR-483 downregulated the expression of PPARa, p21 TIMP2 in HepaRG and HepG2 cells and PPARa and TIMP2 in SK-Hep1 cells (Figure 6D), suggesting that miR-483 inhibits PPARa and TIMP2. The expression of TGFβ also decreased in HepaRG cells after miR-483 overexpression. Collectively our data suggest that targeting PPARa and TIMP2, miR-483 inhibits cell steatosis and fibrosis in HepaRG and HCC cells.

### 3.7. Downregulation of miR-483 Exacerbates NAFLD and AFLD in Mice Liver

Finally, we utilized mouse models of NAFLD and AFLD to understand the biological significance of miR-483 in developing liver disease. C57BL/6J mice were fed either a chow diet (control diet, 12% calories as fat; *n* = 5) or a high-fat diet (HFD, 45% calories as fat; *n* = 5) for 16 weeks as described previously [42] (Figure 7A). Mice fed with HFD showed hepatic steatosis, and this was not observed in the livers of mice fed with the chow diet [42]. We then analyzed the expression of miR-483 in these liver samples (Figure 7B, left panel). RT/qPCR data demonstrated that the expression miR-483 was significantly decreased in the livers of mice fed with HFD. Moreover, we analyzed the expression of fibrosis biomarkers and Notch(s) receptors in mice liver fed with HFD (Figure 7B, right panels, and Appendix A). Mice fed with HFD showed significantly increased expression of *TIMP2*, *TGFβ* (*p* < 0.05), and downregulated *AST* (*p* < 0.001) and *Notch2* (*p* < 0.01) expression in liver tissues. No significant change in *p21*, *Notch1*, *Notch3*, or *Notch4* was detected (Figure 7B, right panels, and Appendix A).

Similarly, we also analyzed the expression of miR-483, fibrosis markers, and Notch(s) receptors in an AFLD mice model (Figure 7C). Mice fed an EtOH-containing diet showed hepatic steatosis and not in mice liver fed with the control diet [42]. Similar to the NAFLD mice model, expression of miR-483 was also downregulated in mice liver fed with EtOH containing diet (*p* < 0.01) compared with control diet-fed animals (Figure 7D, left panel). Fibrogenic *TGFβ* expression was significantly increased (*p* < 0.01) in these samples. However, the expression of *TIMP2* and *AST1* was downregulated. No significant change in the expression of *p21*, *Notch1*, *Notch2, and Notch4* was observed in these samples (Figure 7D, right panels, and Appendix A). Increased expression of Notch3 was detected in mice liver fed with EtOH containing diet (Figure 7D, right panels, and Appendix A). Considering these results, our data suggest that miR-483 expression is downregulated in both NAFLD and AFLD mice models that could be involved in the modulation of the fibrogenic signaling in mice liver.

## 4. Discussion

Hepatocellular carcinoma (HCC) carries a significant threat of cancer-related mortality [1]; thus, identifying biomarkers in the early stages of NAFLD, AFLD, and subsequent HCC progression is crucial. Due to their stability in various bodily fluids, miRNAs are currently studied as prognostic and diagnostic markers for various diseases, including cancer [54]. A recent study showed the under-expression of miR-370-3p and over-expression of miR-196a-5p in serum exosomes of HCC patients compared to control samples. De-regulation of these miRNAs was also associated with increased tumor size, tumor grade, Tumour Node Metastasis (TNM) stage, and worsened prognosis [55]. Another study showed higher expression of circulating miRNAs such as miR-16 and miR-122 in early-stage HCC patients with high diagnostic efficacy [56]. The diagnostic utility of miR-122-5p, miR-21-5p, and miR-222-3p was also analyzed in the serum samples of patients with hepatitis C viral infection and HCC post-direct-acting antiviral (DAAs) therapy [57]. Interestingly, downregulation of miR-21-5p and miR-122-5p was observed in the HCC post-DAA therapy group compared to control samples. In contrast, higher expression of miR-21-5p and miR-122-5p was detected in the HCV-related HCC group indicating altered expression of these miRNAs during DAAs therapy [57]. These studies underscore the prognostic and diagnostic utility of miRNAs in HCC.

In the present study, we investigated the biological significance of miR-483 in NAFLD, AFLD, and HCC. We found that miR-483 plays an important role in NAFLD/AFLD/HCC modulation, and our data shows downregulation of the expression of miR-483 in HCC patients with a diverse racial background compared to healthy controls. Our data is also supported by the TCGA database, demonstrating the downregulation of miR-483-5p and miR-483-3p in HCC. Further, we demonstrated that overexpression of miR-483 inhibited HCC cell survival/migration, increased anti-HCC drug sensitivity, and induced cell apoptosis. Since Notch(s) signaling is constitutively activated in HCC and involved in tumor formation [58], we also analyzed the effect of miR-483 expression regulation in Notch(s) signaling in HCC cells. Our data indicated that overexpression of miR-483 inhibits Notch3 in HepG2 and SK-Hep1 cells, thus downregulating the expression of Notch downstream target HES1. However, the molecular mechanism of how miR-483 dysregulates Notch signaling in HCC remains to be investigated. MicroRNAs play a dual role as a tumor suppressor as well as oncogenes. Numerous studies suggest that under or overexpression of specific miRNA or antagomirs can affect the downstream gene regulatory network/cell signaling pathways, which could lead to reversing the phenotypes in cancer cells [59]. MicroRNAs regulate drug resistance in HCC; for example, the downregulation of miR-122 upregulates ABCB1, ABCF2, and PKM2 and increases resistance against doxorubicin [60], whereas the downregulation of miR-340 activates Nrf2 and enhances resistance against cisplatin [61]. Our data suggest that down regelation miR-483 activates Notch, whereas overexpression of miR-483 inhibits HCC hallmarks and increases sensitivity toward anti-HCC drugs. Analysis of miR-483 expression in the different stages of HCC progression needs to be monitored to control not only HCC progression but also NAFLD and AFLD, and designing strategies for overexpression miR-483 in the liver along with anti-HCC drugs may give a better outcome for NAFLD and AFLD and HCC patients.

Liver cirrhosis is one of the main causes of HCC and can be caused by various underlying etiologies like chronic hepatitis B/C viral infection, NAFLD, and AFLD [3,4,5,6]. López-Riera, et al. recently identified nine serum microRNAs, miR-16, miR-21, miR-22, miR-27b, miR-30c, miR34a, miR-122, miR-192, and miR-197 associated with NAFLD severity [62]. Furthermore, miR-22, miR-27b, miR-192, and miR-197 appeared to be NAFLD-specific compared with drug-induced liver injury [62]. As previously reviewed [63], the expression of miR-27a, miR-140-5p, miR-191, miR-222, miR-224, miR-378a-3p, miR-140-5p, miR-483, and miR-520d-5p modulates pathogenesis of hyperlipidemia by targeting proprotein convertase subtilisin/kexin type 9 (PCSK9). Specifically, miR-191, miR-222, and miR-224 miR-483-5p control PCSK9 expression [34,64,65], enhanced hypercholesterolemia [34], and LDL-C uptake in mice liver fed with a high-fat diet [65]. Our data indicate that HCC cells exposed to different fatty acids, particularly oleic acid (OA), lauric acid (LA), and cholesterol (CHO), significantly increased miR-483 expression in HepaRG and HepG2 cells and overexpression of miR-483 downregulated cell steatosis by suppressing lipogenic gene expression in vitro. Our data also suggest that miR-483 targets PPARa and downregulates cell steatosis and overexpression of miR-483 increased LC3B autophagy biomarker expression since cellular autophagy modulates lipid metabolism [50,66]. These results suggest that mir-483 suppresses cell steatosis by targeting PPARa and by induction of autophagy. 

Hepatic steatosis progresses from fibrosis and cirrhosis to HCC, and earlier studies demonstrated the dysregulation of microRNA expression in these transitions [67,68,69]. For example, higher expression of miR-199 was observed in liver fibrosis [70], but its expression was down-regulated in HCC [71]. Interestingly, the increased expression of miR-483 was detected in advanced cirrhosis patients infected with hepatitis C virus [72], and overexpression of MiR-483 suppresses CCl_4_ mediated induction of fibrosis in mice liver [35], suggesting that miR-483 modulates liver fibrosis. A recent report suggests that under-expression of miR-483 in serum from patients with idiopathic pulmonary arterial hypertension (IPAH) revealed that miR-483 targets several PAH-related genes, including transforming growth factor-β (TGF-β), TGF-β receptor 2 (TGFBR2), β-catenin, connective tissue growth factor (CTGF), interleukin-1β (IL-1β), and endothelin-1 (ET-1) and overexpression of miR-483 in endothelial cells (ECs) inhibited inflammatory and fibrogenic responses [51]. Similarly, we found that overexpression of miR-483, particularly HepG2 and SK-Hep1 cells suppressed the expression of fibrosis markers such as TIMP2, TGF-β, cytoleratin7, and CX3CR1. Increased expression of miR-483 was observed in HepG2 when cells were exposed to CCl_4_, leading to the downregulation of TIMP2 and TGFβ. This suggests that miR-483 suppresses fibrogenic signaling in HCC, as reported earlier [35].

Since miR-483 modulates cell steatosis and fibrogenic signaling in HCC cells, we wanted to investigate further how miR-483 affects these processes. We first analyzed the possible targets of miR-483 that are involved in steatosis and fibrosis. Interestingly our analysis revealed that miR-483 targets the PPARA-3’UTR sequence (although poorly conserved). Our reported data suggest that miR-483 binds to the *PPARA (5)-3’UTR* and affects *PPARA* expression. The PPARs are known to regulate lipid metabolic enzyme expression and modulate intracellular lipid metabolism, entry of fatty acid into peroxisome and mitochondria, and mitochondrial fatty acid catabolism [73]. In the liver, PPARα regulates lipid metabolism and controls liver homeostasis, and dysregulation and overexpression of PPARα may lead to hepatic steatosis, steatohepatitis, steatofibrosis, and liver cancer [53]. An earlier study suggests that miR-483-3p inhibited adipocyte differentiation by reducing the expression of PPARγ2 and FABP4, and miR-483-3p antagonist treatment (9 days) increased the expression of PPARγ2 and FABP4, indicating that miR-483 may dysregulate PPARs expression [74]. Our data also supports the observation that miR-483 inhibits PPARa expression and downregulates cell steatosis.

MiR-483 is known to target tissue inhibitors of metalloproteinases 2 (TIMP2), and a recent study suggests that inhibition of miR-483-5p by intra-articular injection of antago-miR-483-5p could prevent the onset of osteoarthritis (OA) pathogenesis by targeting Matrilin 3 (Matn3) and TIMP2 [75]. Bone marrow mesenchymal stem cells derived exosomal miR-483 increased multiple myeloma’s malignant progression by inhibiting TIMP2 expression [76], and the inhibition of miR483 increased expression of p21 and downregulated the expression of c-Myc and Bcl-2 [76]. Li et al. demonstrated that miR-483 suppresses CCl_4_-mediated mouse liver fibrosis in vivo by targeting TIMP2 and PDGF-β [35]. Our RT/qPCR, immunoblotting, and reporter assay data suggest that miR-483 targets TIMP2 expression and suppresses fibrogenic signaling in HCC cells. Our data also suggest that miR-483 potentially suppressed steatotic and fibrogenic response by targeting PPARA, TIMP2, TGFB1, and p21 since miR-483 binds with the UTRs of these genes.

We further established the association between miR-483 expression and NAFLD and AFLD progression in vivo mice models (Figure 7E). Although an earlier study demonstrated that miR-483-5p targets PCSK9, increases hepatic LDL Receptor expression, and ameliorates hypercholesterolemia in mice liver [34], our data suggest that mice fed with a high-fat diet and EtOH show downregulation of miR-483 expression compared with mice fed a regular diet, similar to our HCC tissue samples. Downregulation of miR-483 in the NAFLD mouse model increased *TIMP2* and *TGF-β* expression, whereas down-regulation of miR-483 in the AFLD mouse model increased *TGF-β* expression but not *TIMP2*, indicating that miR-483/TIMP2 axis differentially regulated in fatty liver disease.

## 5. Conclusions

Our data suggest that the downregulation of miR-483 in NAFLD/AFLD and HCC modulates the progression of fatty liver diseases and HCC, and overexpression of miR-483 inhibited cell steatosis, fibrosis, and HCC cell proliferation; therefore, miR-483 may be used as a novel therapeutic target to treat patients with fatty liver diseases/HCC.

## Figures and Tables

**Figure 1 cancers-15-01715-f001:**
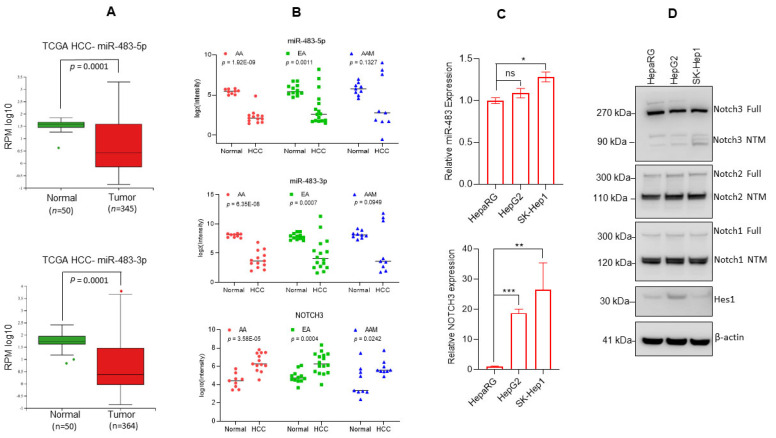
Downregulation of miR-483 in HCC. (**A**, **upper and lower panels**): Expression of miR-483-5p and miR-483-3p as reported in The Cancer Genome Atlas Liver Hepatocellular Carcinoma (TCGA-LIHC) data set is presented (MIR-TV: http://mirtv.ibms.sinica.edu.tw/analysis.php, accessed on 9 August 2022). *p* = 0.0001 compared with normal liver tissues. (**B**) Expression of mir-483-5p, miR-483-3p, and Notch 3 in HCC patient tumors of African American (AA), European American (EA), and Asian American (AAM) was analyzed by RNAseq. *p* values are presented in the graphs compared with normal HCC tissues (**Upper, middle, and lower panels**). (**C**) Endogenous expression of miR-483-5p and *Notch 3* in HepaRG and HCC cells was determined by RT/qPCR. * *p* < 0.05, ** *p* < 0.01, *** *p* < 0.001 compared to HepaRG cells. ns: not significant. (**D**) Cell lysates of HepaRG, HepG2, and SK-Hep1 cells were immunoblotted with anti-Notch 3, anti-Notch 2, anti-Notch 1, Hes1, and β-actin antibodies. The uncropped blots are shown in Appendix A.

**Figure 2 cancers-15-01715-f002:**
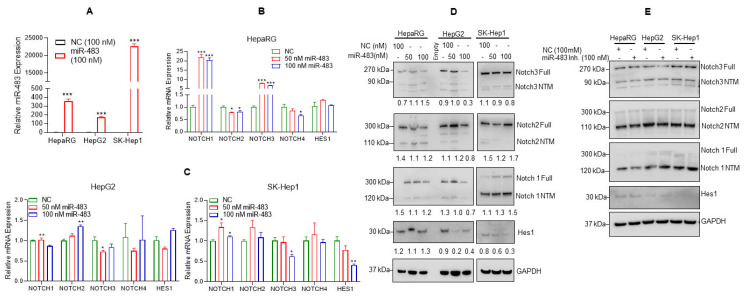
miR-483 affects Notch signaling. (**A**) HepaRG, HepG2, and SK-Hep1 cells were transfected with NC (negative control) mimic or miR-483 mimic for 24 h, and the expression of miR-483 was analyzed by RT/qPCR. *** *p* < 0.001 compared to NC-transfected cells. (**B**,**C**) Expression of *Notch 1*, *Notch 2*, *Notch 3*, *Notch 4,* and *HES 1* mRNA levels after transfection of HepaRG, HepG2, and SK-Hep1 cells with NC and miR-483 mimic was analyzed by RT/qPCR. * *p* < 0.05, ** *p* < 0.01, *** *p* < 0.001 compared to NC transfected cells. (**D**) HepaRG, HepG2, and SK-Hep1 were transfected with NC and miR-483 mimics for 24 h, and cell lysates were immunoblotted with anti-Notch 3, anti-Notch 2, anti-Notch 1, Hes1, and GAPDH antibodies. The band intensities were quantified using ImageJ software (https://imagej.nih.gov/ij/, accessed on 12 February 2023) and presented. (**E**) HepaRG, HepG2, and SK-Hep1 were transfected with NC, or miR-483 inhibitor (Inh.) for 24 h, and cell lysates were immunoblotted with anti-Notch 3, anti-Notch 2, anti-Notch 1, Hes1, and GAPDH antibodies. The uncropped blots are shown in Appendix A.

**Figure 3 cancers-15-01715-f003:**
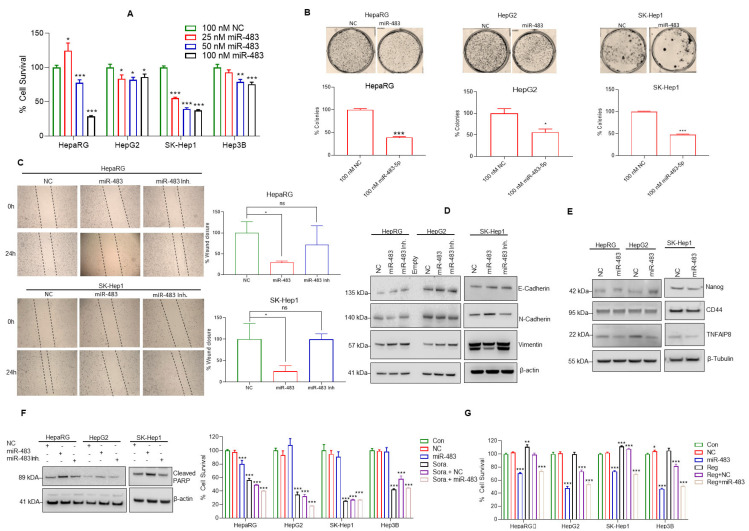
miR-483 inhibits HCC hallmarks. (**A**) HepaRG, HepG2, SK-Hep1, and Hep3B cells were transfected with NC (100 nM) or increasing concentrations of miR-483 mimic (25 to 100 nM) as indicated for 48 h. The effect of miR-483 on HCC cell survival was determined by MTT assay as described in the materials and methods section. * *p* < 0.05, ** *p* < 0.01, *** *p* < 0.001 compared to NC transfected cells. (**B**) The effect of miR-483 on HepaRG and HCC cell colony formation was analyzed as described in the materials and methods section. The number of colonies was quantified and plotted (lower panel). * *p* < 0.05, *** *p* < 0.001 compared to NC transfected cells. (**C**) The effect of miR-483 mimic and miR-483 inhibitor on HepaRG and SK-Hep1 cell migration was analyzed by wound-healing/scratch assay. Representative images of the wound healing assay (left panels) and the percentages of wound closure were determined by ImageJ software and plotted. (Right panels). * *p* < 0.05 compared to NC-transfected cells. ns: not significant. (**D**) HepaRG, HepG2, and SK-Hep1 cells were transfected with NC or miR-483 mimic or miR-483 inhibitor for 24 h, and the expression of E-cadherin, N- Cadherin, Vimentin, and β-actin were analyzed by immunoblotting. (**E**) The effect of miR-483 mimic on the expression of Nanog, CD44, TNFAIP8, and β-tubulin in HepaRG, HepG2, and SK-Hep1 cells was analyzed by immunoblotting. (**F**) HepaRG, HepG2, and SK-Hep1 cells were transfected with NC or miR-483 mimic or miR-483 inhibitor for 24 h, and the expression of cleaved-PARP and β-actin was analyzed by immunoblotting. (**G**) HepaRG, HepG2, SK-Hep1, and Hep3B cells were transfected with NC or miR-483 alone or in combination with sorafenib (5 µM) or regorafenib (2.5 µM) as indicated for 48 h and cell survival was determined by MTT assay (left and right panels). * *p* < 0.05, ** *p* < 0.01, *** *p* < 0.001 compared to the untreated/NC transfected and sorafenib/regorafenib treated cells. The uncropped blots are shown in Appendix A.

**Figure 4 cancers-15-01715-f004:**
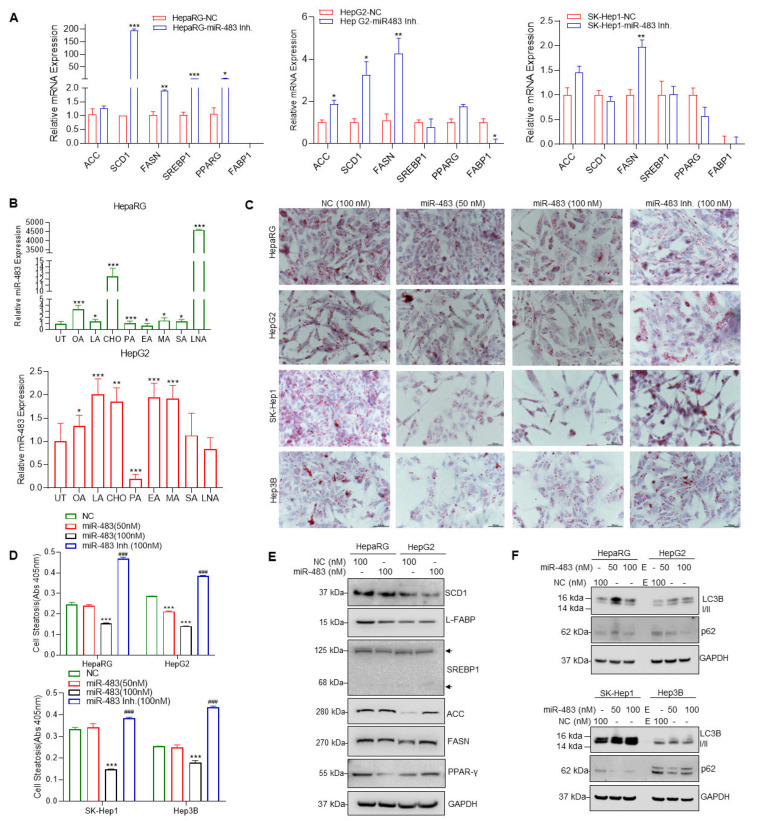
(**A**) HepG2 and SK-Hep1 cells were transfected with NC (100 nM) or miR-483 inhibitor (100 nM) as indicated for 30 h, and the expression of *FASN*, *SREBP1*, *ACC*, *FABP*, *PPARγ,* and *SCD1* were analyzed RT/qPCR. * *p* < 0.05, ** *p* < 0.01, *** *p* < 0.001 compared to NC transfected cells. (**B**) HepaRG and HepG2 cells were treated with 100 µM of indicated fatty acids for 30 h, and the expression miR-483 was analyzed by RT/qPCR. * *p* < 0.05, ** *p* < 0.01, *** *p* < 0.01 compared with untreated cells (upper and lower panels). (**C**) HepaRG and HCC cells were grown on coverslips and transfected with NC (100 nM) or miR-483 mimic (100 nM) or miR-483 inhibitor as indicated for 18 h and treated with oleic acid (100 µM) for an additional 24 h. Cells were fixed, stained with Oil Red O (ORO), and images were captured using a Nikon Y-IDP microscope. (**D**) HepaRG and HCC cells were grown in 6-well plates in triplicates, transfected with NC, miR-483, and miR-483 inhibitor for 18 h, and then treated with 100 µM oleic acid for an additional 24 h. Cells were fixed and stained with an Oil Red O (ORO), cells were lysed, and the Oil Red O stain released from steatotic cells was measured by monitoring the absorbance at 405 nm using a plate reader. *** *p* < 0.001, ^###^
*p* < 0.001 compared with NC-treated cells. (**E**) HepaRG and HepG2 cells were transfected with NC (100 nM) or miR-483 mimic (100 nM) as indicated for 30 h, and the expression of FASN, SREBP1, ACC, L-FABP, PPARγ, SCD1, and GAPDH were analyzed by immunoblotting. (**F**) HepaRG, HepG2, SK-Hep1, and Hep3B cells were transfected with NC or miR-483 as indicated for 30 h, and the expression of LC3B I/II, p62, and GAPDH was analyzed by immunoblotting. The uncropped blots are shown in Appendix A.

**Figure 5 cancers-15-01715-f005:**
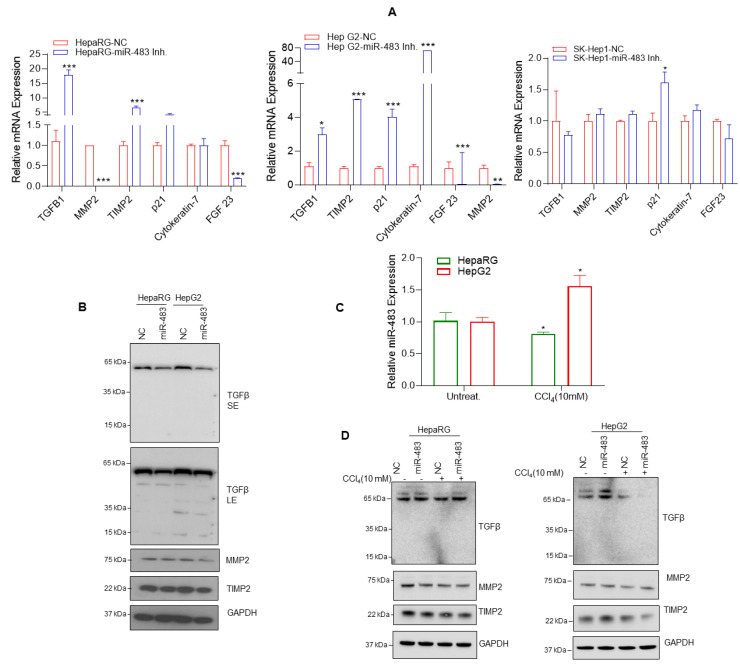
miR-483 modulates fibrosis. (**A**) HepaRG and HCC cells were transfected with NC, or miR-483 inhibitor for 30 h, and the expression of key genes related to the fibrogenic signaling such as *TIMP2*, *MMP2*, *TGF-β*, *Cytokeratin-7*, *p21,* and *FGF-23* were analyzed by RT/qPCR (all panels). * *p* < 0.05, ** *p* < 0.01, *** *p* < 0.001 compared with NC transfected cells. (**B**) HepaRG and HepG2 cells were transfected with NC or miR-483 as indicated for 30 h, and the expression of TGFβ, TIMP2, MMP2, and GAPDH was analyzed by immunoblotting. (**C**) HepaRG and HepG2 cells were treated with vehicle or CCl_4_ as indicated for 72 h, and expression of miR-483 was analyzed by RT/qPCR. * *p* < 0.05 compared with NC-transfected cells. (**D**) HepaRG and HCC cells were transfected with NC or miR-483 inhibitor and treated with CCl_4_ as indicated for 72 h; the expression of TGFβ, TIMP2, MMP2, and GAPDH was analyzed by immunoblotting. The uncropped blots are shown in Appendix A.

**Figure 6 cancers-15-01715-f006:**
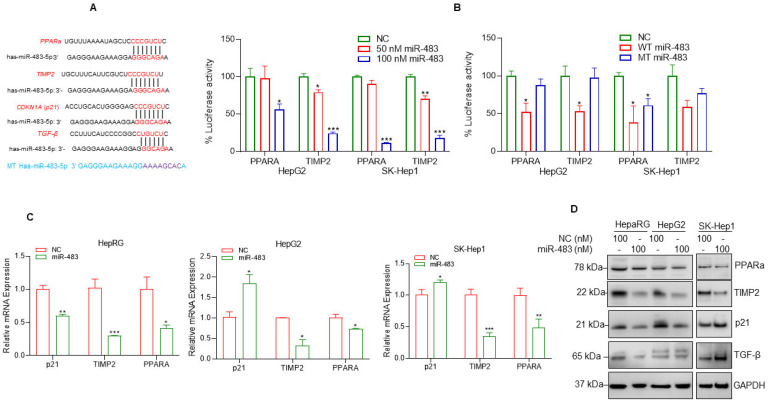
miR-483 targets PPARa and TIMP2 UTRs and inhibits their expression. (**A**) The binding sites of miR-483 in the 3′UTR of *PPARA* (5), *TIMP2,* and *p21* gene were analyzed by TargetScan (http://www.targetscan.org/vert_72/, accessed on 12 August 2022) and presented. The binding site of miR-483 in the 3′UTR of TGFB was also presented. We also generated custom-made wild-type miR-483 and mutated miR-483 nucleotide sequences and presented them. (**B**) Luciferase reporter assay: HCC HepG2 and SK-Hep1 cells were transfected with *PPARA*-3′UTR-Luciferase reporter construct (0.5 µg DNA), or *TIMP2*-3′UTR-Luciferase reporter construct (0.5 µg DNA) for 18 h and then cells were transfected with NC mimic or wild-type-miR-483 (50 nM and 100 nM; left panel) or wild-type-miR-483 or mutant-miR-483 (100 nM; right panel) for an additional 24 h. Transfected cells were lysed, and luciferase activity was measured. Results are representative of three independent experiments. * *p* < 0.05, ** *p* < 0.01, *** *p* < 0.001 compared with NC transfected cells. (**C**) HepaRG, HepG2, and SK-Hep1 cells were transfected with NC or miR-483 mimic (100 nM) for 30 h, and the effect of NC or miR-483 mimic on *p21*, *TIMP2,* and *PPARa* gene expression was analyzed by RT/qPCR. * *p* < 0.05, ** *p* < 0.01, *** *p* < 0.001 compared with NC-transfected cells. (**D**) HepaRG, HepG2, and SK-Hep-1 cells were transfected with NC mimic or miR-483 mimic for 30 h, and expression of endogenous p21, TIMP2, PPARa, and TGFβ protein expression was analyzed by western blotting. The uncropped blots are shown in Appendix A.

**Figure 7 cancers-15-01715-f007:**
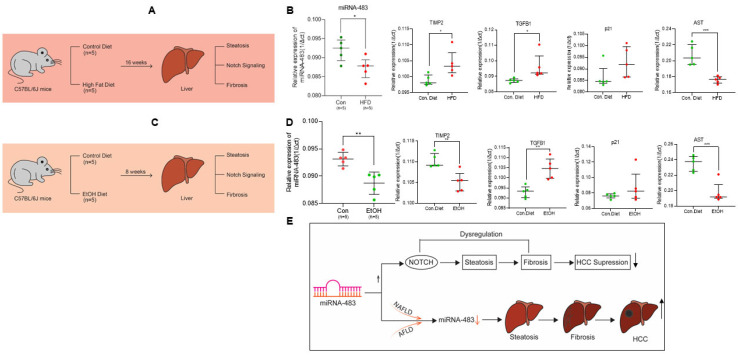
Association of miR-483 and fibrogenic markers expression in NAFLD and AFLD mouse model. (**A**) The schematic represents the strategy for the development of NAFLD as described in the materials and methods section. (**B**) Male C57BL/6J mice pair-fed with a control diet (*n* = 5) or HFD (*n* = 5) for 16 weeks, total RNAs from liver tissues were also isolated, and the expression of miR-483 and fibrosis signaling gene expression markers were analyzed by RT/qPCR (Green circles represent the control mice liver samples, and the red circles represent the HFD-fed mice liver samples). * *p* < 0.05, *** *p* < 0.001 compared with control diet-fed mice. (**C**) The schematic represents the strategy for the development of the AFLD mouse model. (**D**) Male C57BL/6J mice pair-fed with a control diet (*n* = 5) or EtOH diet (*n* = 5) for 8 weeks, total RNAs from liver tissues were isolated, and the expression of miR-483 and fibrosis signaling gene expression markers were analyzed by RT/qPCR (Green circles represent the control mice liver samples, and the red circles represent the EtoH-fed mice liver samples). ** *p* < 0.01, *** *p* < 0.001 compared with control diet-fed mice. (**E**) A schematic model represents the role of miR-483 in the inhibition of HCC cell proliferation and its association with NAFLD and AFLD progression.

## Data Availability

Data supporting reported results can be found at https://www.ncbi.nlm.nih.gov/geo/query/acc.cgi?acc=GSE176289, accessed on 12 August 2022) as indicated.

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
