# Peer review of "MicroRNA-483-5p Inhibits Hepatocellular Carcinoma Cell Proliferation, Cell Steatosis, and Fibrosis by Targeting PPARα and TIMP2"

_cancers, 2023, doi:10.3390/cancers15061715_

Round 1

Reviewer 1 Report

The manuscript entitled "MicroRNA-483-5p inhibits hepatocellular carcinoma cell proliferation, cell steatosis, and fibrosis by targeting PPARα and 3 TIMP2" by Niture et al., is very significant in liver associated diseases. The rationale and hypothesis of this work is strong. All parts of the manuscript are appropriate and lead to be a conclusion. Although, I do not find major issues, but I have few suggestions to offer which may improve quality of the manuscript:

1. What is the motto of Figure 1D? Is it better to include any non-cancer cells for comparison?

2. All western blot color must be turned to gray rather than reddish/pinkish color.

3. How authors confirmed various stages of liver disease (NAFLD, steatosis, fibrosis, HCC) progressing to HCC? Did any markers used in this study is not clear.

Author Response

Reviewer 1: Comments and Suggestions for Authors

The manuscript entitled "MicroRNA-483-5p inhibits hepatocellular carcinoma cell proliferation, cell steatosis, and fibrosis by targeting PPARα and 3 TIMP2" by Niture et al., is very significant in liver associated diseases. The rationale and hypothesis of this work is strong. All parts of the manuscript are appropriate and lead to be a conclusion. Although, I do not find major issues, but I have few suggestions to offer which may improve quality of the manuscript:

Response: Thanks.

  1. What is the motto of Figure 1D? Is it better to include any non-cancer cells for comparison?

Response: In Fig. 1D, we compare the expression of Notch (s) in HepaRG (a human hepatic progenitor cell line that retains many characteristics of primary human hepatocytes) with HCC HepG2 and SK-Hep1 cell lines. We believed that the hepatic stem cell line HepaRG is a well-established in-vitro model more economical and convenient than primary human hepatocytes, so we used it for comparison.   

  1. All western blot color must be turned to gray rather than reddish/pinkish color.

Response: Done.

  1. How authors confirmed various stages of liver disease (NAFLD, steatosis, fibrosis, HCC) progressing to HCC? Did any markers used in this study is not clear?

Response: Using human hepatocytes, HepaRG cells, and HCC cell lines in vitro and using NAFLD and AFLD mouse models in vivo, in this study we established an association of how miR-483 expression correlates with cell steatosis and expression of lipogenic biomarkers (SREPB1 PPARγ, PPARα, FABP1, SCD1, FASN, and ACC). We also analyzed the impact of miR-483 on fibrosis markers (TGFB1, MMP2, TIMP2, p21 Cytokaratin-7 and FGF23).  Also, we analyzed the effect of CCl4-mediated and miR-483-mediated impact on fibrosis markers in HepaRG and HepG2 cell lines.   These steatosis and fibrosis biomarkers are well-established in liver diseases.     

Reviewer 2 Report

It is an interesting manuscript. It is a very well-written paper and only a minor comment to be addressed prior to publication:

1 - Can you please comment if there is any data on lncRNA which can be a competitive endogenous RNA for the mir-483.

2 - Please comment in the discussion how the described mechanism can be related to the therapy of HCC and its potential impact on outcome software chemo/immunotherapy.

Author Response

Reviewer 2: Comments and Suggestions for Authors

It is an interesting manuscript. It is a very well-written paper and only a minor comment to be addressed prior to publication:

Response: Thanks.

1 - Can you please comment if there is any data on lncRNA which can be a competitive endogenous RNA for the mir-483.

Response: This information was added in the introduction section of the revised manuscript to explain the regulation of miR-483 by lncRNA.  

2 - Please comment in the discussion how the described mechanism can be related to the therapy of HCC and its potential impact on outcome software chemo/immunotherapy.

Response: Done pl see the updated discussion section.

Reviewer 3 Report

The study is well organized and provides interesting findings. The discussion section can be improved if the authors speculate more about the possible mechanism of regulation by miR-483. I do not detect mayor concerns but only a few comments which are listed below.

1.     Double-check the estimated new cases and deaths reported on line 37. Those do not match the ones on the provided reference.

2.     The link provided on lines 38-39 needs to be fixed.

3.     The word “binds” is missing on line 53; “that to 3’ untranslated regions (UTRs) of genes and inhibit gene expression [16].” [sic].

Author Response

Reviewer 3: Comments and Suggestions for Authors

The authors investigated the pivotal role of MicroRNA-483-5p on HCC. The topic is very interesting and the presentation is wonderful but I have some issues:

  1. I think there are many causes of HCC such as aflatoxins and there are many modules of HCC. So, I strongly recommend concentrating your study on the HCC only and its mechanisms. You must separate nonalcoholic fatty liver disease and alcoholic fatty liver disease into another paper because it is out of the Cancers journal scope.

Response: We agree with the reviewers’ comments; however, in this work, we aimed to investigate the association and impact of miR-483 expression regulation in NAFLD, AFLD, and HCC utilizing our in-vitro and in-vivo models. Both NAFLD and AFLD progress to HCC through histologically defined stages such as hepatic steatosis, fibrosis, and cirrhosis. To assess our hypothesis, we utilized four HCC cell lines, human hepatocytes, and HepaRG cells (a human hepatic progenitor cell line that retains many characteristics of primary human hepatocytes). Therefore, our data is relevant to the scope of the journal. We agreed that the mechanistic role of miR-483 in NAFLD and AFLD needs further investigation, and currently, we are developing those strategies in our laboratory. 

  1. There is a lack of information regarding HCC and its mechanisms in the introduction and much information about nonalcoholic fatty liver disease. Please, increase information about HCC and its causes, therapeutics, and mechanisms and remove the nonalcoholic fatty liver disease.

Response: Thank you for this observation; we added HCC causes and treatment information in the introduction section.  This information will further solidify our rationale.

Line 81: Please, provide the approval number.

Response: IRB approval number provided in revised manuscript.

  1. Line 86: Please, provide all data bout the patients in the supplementary such as which stage of each patient the expression of MicroRNA-483-5p related to the HCC stage, and the prognosis of the disease and samples of histopathological images used to determine each stage.

Response: We do not have any histopathological images available or specific miRNA-483-5p expression profiles for each patient. However, detailed characteristics of the study cohort is described in the supplementary table 1.

  1. Suggest adding markers of HCC as alpha-fetoprotein, Arginase-1, and Glypican-3 to the analysis of Patient samples to confirm the identity of HCC.

Response: We have information on the AFP (shown in supplementary Table 1). Information on other markers for this cohort are not available.

  1. Line 229: I recommend making a HCC mice module to confirm your hypothesis instead of developing of NALD and AFLD mouse models.

Response: We are currently developing strategies to expand our studies utilizing HCC in-vivo models. However, this model was separate from the original scope of this specific work. Since revising this part of the manuscript and including a new model will take significant time, we will include our current models and keep expanding our studies for future publication.

  1. Figure 7 B and C: the liver of normal control mice contains many necrosis and degeneration suggesting diseases in investigated mice. Please, adjust the laboratory animals' environment to produce healthy normal mice without any lesions in the liver. 

Response: The slides were previously scored by a certified pathologist who did not find necrosis. However, the Lieber-DeCarli control liquid diet we used in this study contained 35% calories from fat, 47% from carbohydrate, and 18% from protein (Dyets Inc, Bethlehem, PA) which may sometimes cause mild lipid accumulation, especially in male mice and not necrosis. Furthermore, the mice were not starved before sacrifice and the presence of glycogen may also be observed in hepatocytes.

Reviewer 4 Report

* The authors investigated the pivotal role of MicroRNA-483-5p on HCC. The topic is very interesting and the presentation is wonderful but I have some issues:

1. I think there are many causes of HCC such as aflatoxins and there are many modules of HCC. So, I strongly recommend concentrating your study on the HCC only and its mechanisms. You must separate nonalcoholic fatty liver disease and alcoholic fatty liver disease into another paper because it is out of the Cancers journal scope.

2. There is a lack of information regarding HCC and its mechanisms in the introduction and much information about nonalcoholic fatty liver disease. Please, increase information about HCC and its causes, therapeutics, and mechanisms and remove the nonalcoholic fatty liver disease.

3. Line 81: Please, provide the approval number.

4. Line 86: Please, provide all data bout the patients in the supplementary such as which stage of each patient,  the expression of MicroRNA-483-5p related to the HCC stage, and the prognosis of the disease and samples of histopathological images used to determine each stage.

5.  Suggest adding markers of HCC as alpha-fetoprotein, Arginase-1, and Glypican-3 to the analysis of Patient samples to confirm the identity of HCC.

6. Line 229: I recommend making a HCC mice module to confirm your hypothesis instead of developing of NALD and AFLD mouse models.

7. Figure 7 B and C: the liver of normal control mice contains many necrosis and degeneration suggesting diseases in investigated mice. Please, adjust the laboratory animals' environment to produce healthy normal mice without any lesions in the liver. 

Author Response

Reviewer 4: Comments and Suggestions for Authors

The study is well organized and provides interesting findings. The discussion section can be improved if the authors speculate more about the possible mechanism of regulation by miR-483. I do not detect major concerns but only a few comments which are listed below.

Response: Thank you.

  1. Double-check the estimated new cases and deaths reported on line 37. Those do not match the ones on the provided reference.

Response: We verified this and assured it matches the reference provided.

  1. The link provided on lines 38-39 needs to be fixed.

Response: Fixed.

  1. The word “binds” is missing on line 53; “that to 3’ untranslated regions (UTRs) of genes and inhibit gene expression [16].” [sic].

Response: Corrected.

Round 2

Reviewer 4 Report

I have been satisfied with their comments. However, the in vivo negative control samples clearly revealed pathologic lesions by histopathologic examination. I recommend accepting this manuscript after removing this doubting part.

Author Response

I have been satisfied with their comments. However, the in vivo negative control samples clearly revealed pathologic lesions by histopathologic examination. I recommend accepting this manuscript after removing this doubting part.

Response:  Thanks. We modified Fig. 7  by removing the liver steatosis pictures and related text in the revised manuscript.